# Species-Specific *N*-Glycomes and Methylation Patterns of Oysters *Crassostrea gigas* and *Ostrea edulis* and Their Possible Consequences for the Norovirus–HBGA Interaction

**DOI:** 10.3390/md21060342

**Published:** 2023-06-02

**Authors:** Audrey Auger, Shin-Yi Yu, Shih-Yun Guu, Agnès Quéméner, Gabriel Euller-Nicolas, Hiromune Ando, Marion Desdouits, Françoise S. Le Guyader, Kay-Hooi Khoo, Jacques Le Pendu, Frederic Chirat, Yann Guerardel

**Affiliations:** 1Univ. Lille, CNRS, UMR 8576-UGSF-Unité de Glycobiologie Structurale et Fonctionnelle, F-59000 Lille, France; audrey.auger.etu@univ-lille.fr (A.A.); shinyi.yu@gmail.com (S.-Y.Y.); frederic.chirat@univ-lille.fr (F.C.); 2Institute of Biological Chemistry, Academia Sinica, Nangang, Taipei 11529, Taiwan; ufogirl100@gmail.com (S.-Y.G.); kkhoo@gate.sinica.edu.tw (K.-H.K.); 3Nantes Université, Inserm UMR 1307, CNRS UMR 6075, Université d’Angers, CRCI2NA, F-44000 Nantes, France; agnes.quemener@univ-nantes.fr; 4MASAE Microbiologie Aliment Santé Environnement, Ifremer, BP 21105, 44311 Nantes, France; gabriel.euller@ifremer.fr (G.E.-N.); marion.desdouits@ifremer.fr (M.D.); soizick.le.guyader@ifremer.fr (F.S.L.G.); 5Institute for Glyco-core Research (iGCORE), Gifu University, Gifu 501-1193, Japan; ando.hiromune.i0@f.gifu-u.ac.jp; 6Institute of Biochemical Sciences, National Taiwan University, Taipei 10617, Taiwan; 7Immunology and New Concepts in ImmunoTherapy, Nantes Université, Inserm, CNRS, UMR 1302/EMR6001, 44200 Nantes, France; jacques.le-pendu@univ-nantes.fr

**Keywords:** glycomics, norovirus ligands, oysters, methylation

## Abstract

Noroviruses, the major cause of acute viral gastroenteritis, are known to bind to histo-blood group antigens (HBGAs), including ABH groups and Lewis-type epitopes, which decorate the surface of erythrocytes and epithelial cells of their host tissues. The biosynthesis of these antigens is controlled by several glycosyltransferases, the distribution and expression of which varies between tissues and individuals. The use of HBGAs as ligands by viruses is not limited to humans, as many animal species, including oysters, which synthesize similar glycan epitopes that act as a gateway for viruses, become vectors for viral infection in humans. Here, we show that different oyster species synthesize a wide range of *N*-glycans that share histo-blood A-antigens but differ in the expression of other terminal antigens and in their modification by *O*-methyl groups. In particular, we show that the *N*-glycans isolated from *Crassostrea gigas* and *Ostrea edulis* exhibit exquisite methylation patterns in their terminal *N*-acetylgalactosamine and fucose residues in terms of position and number, adding another layer of complexity to the post-translational glycosylation modifications of glycoproteins. Furthermore, modeling of the interactions between norovirus capsid proteins and carbohydrate ligands strongly suggests that methylation has the potential to fine-tune the recognition events of oysters by virus particles.

## 1. Introduction

Acute gastroenteritis is the second leading cause of morbidity and mortality in infants and young children worldwide [1]. These infections are very short (about 2 days) and cause diarrhea, vomiting and fever that lead to significant dehydration in young and elderly subjects and may lead to death with more than 500,000 victims per year. In addition to its high social cost, infectious gastroenteritis has a very high economic cost due to care and lost workdays. Among the many causative agents, viruses are responsible for nearly three out of four infections during seasonal gastroenteritis episodes, particularly noroviruses (NoVs) [1,2,3,4]. NoVs are non-enveloped, single-stranded RNA viruses belonging to the *Caliciviridae* family [5]. Among the ten described genogroups, which can be broken down into numerous genotypes, only five have been described to be involved in human pathologies (GI, GII, GIV, GVIII and GIX) with a high prevalence (80 to 90% of cases) for the GII strain [6]. NoV genotypes responsible for human infections are known to recognize histo-blood group antigen (HBGA) glycan motifs including ABH and Lewis (Le) antigens decorating the surface glycans of human erythrocytes and epithelial cells of different tissues [7]. They are also present in body fluids (serum, saliva, milk, etc.) either associated with proteins or as free oligosaccharides [8]. The biosynthesis of these antigens is under the control of several glycosyltransferases, which are differentially distributed and expressed in different tissues and individuals [9] and which act on multiple carbohydrate substrates [10]. For glycoproteins, the synthesis is initiated by the addition of an α1-2Fuc residue on the Gal residue of type-1 or type-2 LacNAc motif to yield H antigen. This step is catalyzed by an α1-2 fucosyltransferase encoded by the genes *FUT1* in erythrocytes and *FUT2* in epithelial cells. About 20% of European individuals, described as non-secretor, present an inactive *FUT2* gene and therefore cannot synthesize ABH antigens in epithelia. As a result, they carry only Le^a^ and Le^x^ antigens on the surface of their epithelial cells. Three other glycosyltransferases, namely the α1-3/4 fucosyltransferase encoded by the FUT3 gene, the β1-3 galactosyltransferase B-enzyme and/or the β1-3 *N*-acetylgalactosyltransferase A-enzyme can sequentially or differentially act to generate multiple HBGAs including, A-type 1/2, B-type 1/2, ALe^b/y^ and BLe^b/y^ Ag [10]. Of particular importance, these so-called histo-blood group antigens are also shared with many other animal species, including dogs, pigs and cattle, and, therefore, could contribute to cross-species transmission and zoonotic emergence of NoVs [11,12,13]. Importantly, the recognition of HBGA motifs by enteric viruses is strain-specific, leading to differential susceptibility according to the expression of individual HBGAs [14].

There are multiple routes for the contamination of human food. In particular, oysters are a common vector due to multiple factors that can be summarized as follows [15]. First, oyster production sites, located in coastal waters potentially close to contaminated effluents, are particularly exposed to circular contamination from humans and livestock sewage [6,16,17]. They are also often consumed raw, so unlike most other potentially contaminated seafood, they do not undergo possible heat inactivation. Then, the oyster’s feeding method of filtering large volumes of water allows it to concentrate and accumulate viruses and other pathogens from the environment, even in seawater. [18,19]. Finally, oysters are capable of retaining viruses over a very long time period, up to 4 weeks, despite the depuration procedure before consumption [20,21,22,23], which strongly suggests the occurrence of a specific and high-affinity retention mechanism. Importantly, oyster tissues, including the mantle, digestive tract and gills, were shown to express HBGA-like motifs that may be similar to some of those present in human tissues to which viruses could attach. This would provide a strong rationale for the specific and high-affinity binding of norovirus to oyster tissues [24,25,26]. Consistent with this hypothesis, it was observed that the recognition of these motifs by different virus strains was tissue-dependent. [27,28]. Thus, GI strains bind more strongly to oyster digestive tissues via HBGA-like structures than GII strains, which also bind to the gills and mantle. While these glycans are recognized by HBGA-specific lectins or anti-HBGA [24,26], their exact structure remains unknown, hampering our understanding of the possible selection operated by oysters among the very diverse strains of noroviruses.

Invertebrate *N*-glycans significantly differ from canonical *N*-glycans observed in vertebrates [29]. First, they tend to be shorter with a high proportion of paucimannosylated or pseudohybrid type *N*-glycans. Second, they are substituted by monosaccharides rarely observed in vertebrates such as pentoses (e.g., xylose). Third, the reducing terminal GlcNAc residue is commonly substituted by Fuc residues at both the C-3 and C-6 positions or by a Galβ1-4Fuc disaccharide at C-6. Finally, substituents such as methyl, *N*-methyl-aminoethylphosphonate or phosphoethanolamine groups may also be present. Very little information is currently available on the *N*-glycome of oysters, except for a comprehensive analysis of the *N*-glycans associated with total plasma and hemocytes of the Pacific oyster *Crassostrea virginica*. [30]. In this report, the authors successfully identified a highly variable *N*-glycome characterized by the presence of human histo-blood group A antigen, most of which were methylated on the terminal GalNAc residue, and by a high degree of sulfation on Gal residues. In parallel, a detailed analysis of the binding specificity of *C. virginica* Galectin 1 (CvGal1) showed that it recognized type 2, then type 1, blood group A oligosaccharides with high-affinity and could bind to endogenous hemocyte glycoproteins (β-integrin and dominin) in a carbohydrate-dependent manner [31].

In the present study, we deciphered the tissue-specific *N*-glycomes of two oyster species, *Crassostrea gigas* and *Ostrea edulis* and showed that they synthesize related but significantly different sets of *N*-glycans. Although they partially shared similar terminal HBGA-like antigens, these were substituted on different monosaccharides and at different positions by one to three methyl groups, dramatically increasing the number of exposed antigens. Finally, modeling of the interactions of differentially methylated HBGAs identified in the two species with P-domains of prototypical noroviruses strongly suggests that methylation of carbohydrate ligands may represent a new paradigm for the fine-tuning of carbohydrate–protein interactions in nature and a new molecular player in the diversification of glycans involved the evolution of long-term host–pathogen recognition [32].

## 2. Results

### 2.1. Release and Analysis of N-Glycans

Three organs, namely the gills, digestive tract and mantle, were collected from three separate batches of ten–fifteen individuals from two different species of oysters *C. gigas* and *O. edulis*, and immediately processed according to a standard procedure for N-linked glycans isolation [33]. Briefly, whole proteins and glycoproteins were solubilized from tissues using ultrasound treatment in the presence of triton X100. After removing insoluble materials using centrifugation, the glycoproteins were reduced and alkylated prior to enzymatic cleavage with trypsin in order to facilitate the action of peptide-N-glycosidases (PNGases) used to release *N*-glycans from asparagine residues. *N*-glycans were sequentially released using two PNGases, i.e., PNGase F [34] then PNGase A [35], generating two pools of N-glycans according to the presence or not of C3 substitution, generally by a Fuc residue, of the reducing GlcNAc of the chitobiose core. The two sub-glycomes, i.e., PNGase F and PNGase A, were analyzed individually in order to obtain a more complete interpretation of the spectra.

The analytical workflow for the purified glycans is summarized in Figure 1. In brief, the organ-specific *N*-glycomes in the two oyster species were first analyzed using GC/MS in order to determine their monosaccharide compositions. The *N*-glycan diversity was then evaluated using mass spectrometry in permethylated forms using MALDI-MS and MS^n^ in order to achieve a higher sensitivity and generate more informative MS^n^ spectra in terms of sequence and monosaccharide linkages. The in-depth mapping and interspecies comparison of the natural methylation profiles of *N*-glycans was performed using MS^2^ and MS^3^ sequencing following I-CD_3_ derivation. Finally, individual epitopes were relatively quantified using LC-MS^3^ analyses.

### 2.2. Monosaccharide Composition of N-Glycans

The monosaccharide composition of PNGase F- and PNGase A-released *N*-glycans from both oyster species were established as acetylated alditol residues using GC/MS in comparison with authentic standards of common monosaccharides. As expected, chromatograms showed the presence of Fuc, Xyl, Man, Gal, GalNAc and GlcNAc for glycans isolated from both species, albeit with a much lower proportion of Xyl in *O. edulis* in comparison with *C. gigas* (Figure 2A and Appendix A). Glc was also observed but considered a ubiquitous contaminant that may originate from remnant glycogen that is synthesized in large amounts by oysters [36]. The chromatograms revealed three additional signals at RT 18.8, 26.0 and 30.4 min that did not match with authentic standards (Figure 2A). EI-MS analysis of these three signals typified them as *O*-methylated monosaccharides Me-*O*-deHex (RT 18.8 min), Me-*O*-Hex (TR 26.0 min) and Me-*O*-HexNAc (RT 30.4 min) as demonstrated below. The nature of these monosaccharides and the localization of methyl groups were established using GC/EI-MS following reduction with sodium borodeuteride (NaBD_4_) that allows for discriminating C1 from C6. Under these conditions, the cleavage between skeletal carbons preferentially occurs between a carbon carrying a methoxy group and a carbon carrying an acetyl group, generating two intense fragments, which allows easy positioning of the unique methyl group. In particular, EI-MS analysis of the GC/MS signal at RT 18.8 min identified that *C. gigas* generated two intense primary fragments, [CDHOAc-CHOAc-CHOMe]^+^ at *m*/*z* 190 and [CHOMe-CHOAc-CHOAc-CH_3_]^+^ at *m*/*z* 203. This was confirmed with the observation of secondary fragment ions generated by Mc-Lafferty rearrangements at 130 and 141–101, respectively, which localized the methyl group at the C3 position of the deHex residue and established the presence of Me-3*O*-deHex in *C. gigas* (Figure 2B). EI-MS fragmentation of the signal at the same retention time from *O. edulis* samples generated a very different pattern characterized by two intense fragment ions, [CDHOAc-(CHOAc)_2_-CHOMe]^+^ at *m*/*z* 262 and [CHOMe-CHOAc-CH_3_]^+^ at *m*/*z* 131, which indicated that the deHex residue was methylated on C4 in *O. edulis* (Me-4*O*-deHex) rather than on C3 as in *C. gigas* (Figure 2C). Then, EI-MS analysis of the second signal at RT 26 min showed two major signals at *m*/*z* 190 and *m*/*z* 261 identified as [CHOMe-CHOAc-CDHOAc]^+^ and [CHOAc-CHOAc-CHOAc-CHOMe]^+^ fragments, respectively, indicative of the presence of a Hex residue substituted by a methyl group in C3 position (Me-3*O*-Hex) (Figure 2B). A set of signals at *m*/*z* 189 and *m*/*z* 262, attributed to [CHOAc-CHO-Ac]^+^ and [CHOMe-CHOAc-CHOAc-CDHOAc]^+^ fragments, demonstrated the presence of a very minor methylated hexose at C4 position (Me-4*O*-Hex) co-eluting with the first one. Finally, the signal at RT 30,4 min was characterized as a Me-3*O*-HexNAc residue based on the observation of the primary fragment [CDHOAc-CHNH_2_OAc-CHOMe]^+^ at *m*/*z* 189 and [CHOMe-(CHOAc)_2_-CH_2_OAc]^+^ at *m*/*z* 261. Any possible ambiguity between methylation on the C3 or C4 position due to the combined presence of the deuterium atom on the C1 position and the *N*-acetyl group on the C2 position was lifted by running a similar experiment after NaBH_4_ reduction rather than NaBD_4_.

In the absence of authentic standards to identify the nature of methylated deHex, Hex and HexNAc residues, samples were hydrolyzed, reduced and permethylated prior to analysis, in order to negate the effect of natural methylation on monosaccharide migration, and then compared with authentic reduced permethylated samples (Appendix A). Identification of Fuc as the only deHex signal in all samples established that the presence of Me-3*O*-Fuc and Me-4*O*-Fuc were differentially expressed in *C. gigas* and *O. edulis*. Then, the modification to the relative intensities of permethylated monosaccharides compared to peracetylated ones established the presence of Me-3*O*-GalNAc as well as a mixture of Me-4*O*-Man and Me-3*O*-Man in both oyster species. Altogether, GC-MS analyses of permethylated- and peracetylated-reduced monosaccharides in *N*-glycans isolated from *C. gigas* and *O. edulis* permitted the identification of Me-3*O*-Fuc, Me-4*O*-Fuc, Fuc, Xyl, Me-3*O*-Man, Me-4*O*-Man, Gal, Me-3*O*-GalNAc, GalNAc and GlcNAc as the major components in *N*-glycans (Figure 2A). A very small amount of Me-3*O*-Gal was also found in all of the samples. The identification of methylated monosaccharides in both species of oysters is in total agreement with numerous studies that have reported the occurrence of methylated monosaccharides in invertebrates glycans, including methylated GalNAc and Man residues substituting *N*-glycans isolated from the hemolymph of the *C.virginica* oyster [30]. A comparison of the relative quantities of individual monosaccharides in all twelve *N*-glycan fractions showed striking differences that were more pronounced between species than between tissues (Figure 2D). The more obvious qualitative difference concerned the exclusive presence of Me-3*O*-Fuc in *C. gigas* compared to the presence of Me-4*O*-Fuc in *O. edulis*. Another dramatic difference concerned the much higher prevalence of Xyl in *C. gigas N*-glycans (up to 3.4% in PNGaseF and 9.4% in PNGaseA) compared to *O. edulis* (up to 0.4% in PNGase F and 2.2% in PNGase A) (Appendix A). In contrast, Me-3*O*-Man and Me-4*O*-Man were identified in both species, but at very different ratios depending on the tissue, as illustrated by the large prevalence of Me-3*O*-Man in the gills and mantle compared to the prevalence of Me-4*O*-Man in digestive tissues (Figure 2E). Although the figures are different, a similar trend is observed in both species, except for the *O. edulis* PNGase-A fraction from the gills, in which the proportion of Me-4*O*-Man is slightly higher (Appendix A). Finally, the quantitative composition analysis suggested that *O. edulis* tissues are substituted with a much higher quantity of *N*-glycans than *C. gigas*, although a wide variability was observed between individual organs and the types of *N*-glycans (Appendix A). However, it should be noted that the digestive tract in *C. gigas* is noticeably richer in *N*-glycans than the two other organs, whereas it is lower in *O. edulis*.

### 2.3. N-Glycome Analysis

Permethylated *N*-glycans were detected as their [M+Na]^+^ adducts and their monosaccharide compositions were inferred by calculating the number of Hex, HexNAc, deHex and Pent residues. The nature of the monosaccharides was postulated from the known conserved *N*-glycans biosynthetic pathways and further confirmed using composition analysis. The structural assignment of all glycans was mostly based on the specific CID MS/MS fragmentation patterns afforded by the [M+Na]^+^ molecular ions of permethylated glycans in positive mode, as previously documented [37]. MALDI-QIT-TOF mainly generated sodiated B-fragment ions from the non-reducing end of permethylated *N*-glycans [38]. These B-ions are particularly useful for revealing the presence of antigen motifs such as HBGAs or Lewis antigens. Other relevant B/Y fragmentation events also occurred between the two GlcNAc residues in the chitobiose moiety, which generated B/Y-ions used to identify substituents on the terminal reducing GlcNAc such as common corefucosylation or other substitutions. Moreover, the number of free -OH groups on the terminal Man residues of the core generated with the release of antennae informed the number of antennae. Along the B/Y cleavage events, C/Z ions between Hex and Fuc α/β1-3 HexNAc from the non-reducing end of antennae permitted us to unambiguously distinguish type-1 from type-2 LacNAc antennae and consequently Le^a^/Le^b^ from Le^x^/Le^y^ units. It should be noted that low-energy collisions allowed by this mass spectrometer did not generate intense cross-ring cleavages preventing further information on the bonds.

#### 2.3.1. *N*-Glycome of *C. gigas*

Altogether, using mass spectrometry, we confidently identified 88 *N*-glycans from the *C.gigas* two sub-*N*-glycomes, as illustrated in Figure 3, that were labeled CG1 to CG88 and compiled in Appendix A. These were assigned to four different families, namely paucimannose (26 NG), oligomannose (9 NG), hybrid (2 NG) and complex (51 NG). Out of the 51 complex *N*-glycans, it is noteworthy that the majority of the glycans (34 glycans) were substituted with a single complex antenna that is referred to as pseudo-hybrid according to the recent review by Paschinger and Wilson, whereas the 17 remaining glycans were complex bi-antennary *N*-glycans [29].

Independently of the *N*-glycan types, 69 out of the 88 *N*-glycans isolated from *C. gigas* were substituted on their Man_3_GlcNAc_2_ core either by a Xyl residue (37 *N*-glycans) on the internal Man residue or by one Fuc residue (36 *N*-glycans), two Fuc residues (14 *N*-glycans), one Fuc and one Hex residue (5 *N*-glycans) or a disaccharide Hex_1_Fuc_1_ (4 *N*-glycans) on the reducing GlcNAc residue. Substitution of the terminal-reducing GlcNAc residue by a fucose residue was characterized by B-ions at *m*/*z* M+Na-451, resulting from the chitobiose fragmentation and resulting multiple fragmentations. Core α1-3Fuc and α1-6Fuc were differentiated according to their susceptibility to PNGase F and A. Some *N*-glycans showed a reducing GlcNAc simultaneously substituted by a Fuc and a Hex residue (CG10, 16, 22, 30 and 66), as characterized by the Y-ion at *m*/*z* 678 and its corresponding B-ion (Figure 4A). Substitution of the terminal reducing GlcNAc residue by two fucose residues (CG 6, 13, 17, 28, 31, 36, 43, 46, 51, 62, 71, 72, 73, 77) was also characterized by the Y-ion [Fuc_2_HexNAc] detected at *m*/*z* 648 and its corresponding B-ions at M+Na-625 (Figure 4B). Finally, three glycans (CG20, 32, 65 and 76) with a terminal Fuc_2_HexGlcNAc tetrasaccharide were identified based on the observation of B/Y fragment ions at *m*/*z* 895/852. The observation of an additional Z-ion at parent ion-206 strongly suggested that reducing GlcNAc was substituted by a Fuc residue on the C3 position and by a Fuc-Hex disaccharide on the C6 position, as exemplified for CG32 (Figure 4C). Since CG 20, 65 and 76 were identified from the pool of *N*-glycans released with PNGase F, it was concluded that this disaccharide was bound to the terminal reducing GlcNAc via an α1-6 linkage.

Paucimannose *N*-glycans corresponding to the minimal structure Man_2-4_-HexNAc_2_ were differentiated by the number of their Man residues, the presence of Fuc (0, 1, 2) or Hex (0, 1) residues on the reducing GlcNAc residue and Xyl on the internal Man residue. The presence and positioning of Fuc and Hex residues on the internal GlcNAc residue were easily assessed owing to MS^2^ B-ion fragments resulting from the cleavage of the β1-4 GlcNAc bond of the chitobiose for oligomannosylated *N*-glycans. Depending on the presence of these residues on the Man_2-4_GlcNAc_1_, core fragmentation generated B-ions at *m*/*z* 690 for Man_2_ (CG1, 2, 6, 10), *m*/*z* 850 for Man_2_Xyl (CG4, 5, 13,16), *m*/*z* 894 for Man_3_ (CG3, 8, 9, 17, 20, 22 and 32), *m*/*z* 1054 for Man_3_Xyl (CG7, 14, 15, 28, 29 and 30), *m*/*z* 1098 for Man_4_ (CG11, 19, 21 and 31) or *m*/*z* 1258 (CG18) Man_4_Xyl (Figure 4A).

Out of the nine identified oligomannosidic type *N*-glycans, five signals at *m*/*z* 1580 (CG24), 1784 (CG38), 1988 (CG47), 2192 (CG52) and 2396 (CG63) were assigned to the so-called Man_5_–Man_9_ high-mannose (Man)-type NGs. The signal at *m*/*z* 2600 (CG74), calculated as Hex_10_HexNAc_2_, was also tentatively assigned to the biosynthetic intermediate Glc_1_Man_9_GlcNAc_2_ *N*-glycan. The MS/MS fragmentation patterns of these six compounds did not reveal any obvious deviation from the canonical structures and, thus, these six NGs will not be discussed further. Finally, the signals at *m*/*z* 1754 (CG35), 1914 (CG42) and 2118 (CG50) were assigned to Fuc_1_Man_5_GlcNAc_2_, Fuc_1_Xyl_1_Man_5_GlcNAc_2_ and Fuc_1_Man_6_GlcNAc_2_, respectively, in which Fuc and Xyl substituted internal GlcNAc and internal Man residues owing to core B-fragments at *m*/*z* 1303, 1463 and 1667, respectively.

Two *N*-glycans were classified as hybrid structures (CG41 and 58) for which Manα1-6 and Manα1-3 in the core were substituted by at least one Man residue and one GlcNAc residue, respectively. Their MS/MS patterns were characterized by the release of the LacNAc antennae as a Y-ion at *m*/*z* 1362, identified as [HO-Hex_4_HexNac_2_+Na]^+^ (Figure 4D). Single antennae complex-type *N*-glycans were the most abundant family in *C. gigas* with 34 identified *N*-glycans members (CG12, 23, 25–27, 33–34, 36–37, 39–40, 43–46, 48–49, 51, 53–57, 60–62, 64–66, 70–73 and 77). They were characterized by the absence of substitution on one of the two terminal Man residues in the core, whereas the second one was substituted by at least one GlcNAc residue. The canonical glycosylation pathways suggested that the antennas are associated with Manα1-3 in the penta-saccharide core, but their exact localization on either Manα1-3 or Manα1-6 was not confirmed [40]. The GlcNAcα1-2Man linkages were easily fragmented during the MS^2^ process to yield B- and Y-ions, which corresponded to the terminal non-reducing fragment and the core fragment, respectively. Y-ions generated from pseudohybrid *N*-glycans demonstrated the presence of a single OH-Man, confirming that only one Man residue in the core was substituted. The primary derived Y-ions were detected at either *m*/*z* 1157 for unsubstituted cores (CG 12, 27, 48 and 49), *m*/*z* 1317 for xylosylated cores (CG23, 37, 53 and 56), *m*/*z* 1332 for mono-fucosylated cores (CG25, 26, 39, 40, 54, 55, 57 and 61), *m*/*z* 1492 for xylosylated and mono-fucosylated cores (CG33, 34, 44, 45, 60, 64 and 70), *m*/*z* 1506 for di-fucosylated cores (CG36, 46, 62 and 72) (Figure 4B), *m*/*z* 1536 for Fuc and Hex containing cores (CG65 and 66) and finally *m*/*z* 1666 for xylosylated and di-fucosylated cores (CG43, 51, 71 and 73) (Figure 4E). When present, secondary fragments from Y-ions at *m*/*z* 880 or *m*/*z* 1040 confirmed un-substituted (Hex_3_GlcNAc) (Figure 4B) or Xylose-substituted cores (XylHex_3_GlcNAc) (Figure 4E), respectively.

Seventeen bi-antennary complex-type *N*-glycans were observed in *C. gigas* (CG59, 67, 68, 69, 75, 76, 78–88) and identified according to MS^2^-generated B- and Y-ions generated by the preferential cleavage of GlcNAcβ1-2Man linkage. As mentioned previously, when present, secondary Y-ions generated by the fragmentation of the chitobiose moiety produced signals at *m*/*z* 1026 or *m*/*z* 866 according to the presence or absence of a Xyl residue on the internal Man residue (Figure 4F). Accordingly, Y-ions were detected either at *m*/*z* 1144 for unsubstituted cores (CG59, 69, 79 and 80), *m*/*z* 1304 for Xyl cores (GC67, 75, 81 and 83), *m*/*z* 1318 for Fuc cores (CG68, 82, 85 and 87), *m*/*z* 1478 for XylFuc cores (CG84, 86 and 88) (Figure 4F), *m*/*z* 1492 for Fuc_2_ cores (CG78) or *m*/*z* 1522 for FucHex cores (CG76).

The hybrid- and complex-type *N*-glycans were all characterized by the presence of complex antennae made of different epitopes including di-saccharide mono-LacNAc, tetra-saccharide di-LacNAc, tetra-saccharide A-group and penta-saccharide Le^b^/A-group epitopes that were identified using a combination of MS fragmentation, composition and linkage analyses. Twelve *N*-glycans (CG27, 37, 39–41, 44–46, 51, 69 and 75) were shown to be substituted by one, but never two, LacNAc epitopes, owing to the presence of a B-ion at *m*/*z* 486 and its complementary M-463 Y-ions (Figure 4E). The systematic observation of single- and multiple-fragmentation Z-ions resulting from the cleavage between Gal-GlcNAc strongly suggested the presence of a β1-3 linkage and thus type-1 antennae. Type-1 LacNAc antennae were shown to be further substituted by an additional LacNAc repeat (CG49, 56, 57, 64, 80 and 83), as indicated by the presence of a HexNAc_2_Hex_2_ B-ion at *m*/*z* 935 and a HexNAc_1_Hex_2_ C-ion at *m*/*z* 708 on the corresponding spectra (Figure 4G). The absence of a Z-ion resulting from the cleavage between the terminal Gal residue and the GlcNAc strongly suggested that the additional LacNAc motif was of type-2 rather than type-1.

The vast majority of hybrid- and complex-type *N*-glycans were substituted by a single (CG CG48, 53–55, 58–62, 64–66, 70–73, 75–78, 80 and 83) or two (CG79, 81–82 and 84–88) terminal HexNAc(deHex)Hex trisaccharides, tentatively identified as type-1 A-group epitope GalNAcβ1-3(Fucα1-2)Galβ based on the observation of HexNAc_2_HexdeHex B-ion at *m*/*z* 905 (Figure 4F) and HexNAcHexdeHex C-ion at *m*/*z* 678 (Figure 4H), as well as multiple Y-ions and Z-ions generated from the loss of terminal tetra- and trisaccharides, respectively. The MS^3^ analyses of the C-ion at *m*/*z* 678, as observed in all *N*-glycans, showed B-ions at *m*/*z* 282, in agreement with the presence of the terminal GalNAc of the A-group epitope. Furthermore, the observation of a ^0,2^X-ion at *m*/*z* 271 typified the internal Fucα1-2Gal motif, whereas the ^0,3^X-ion signal at *m*/*z* 301 typified the GalNAcβ1,3Gal motif of the A-group epitope, confirming the attribution of the terminal tri-saccharide as A-group epitope (Figure 4I), in agreement with monosaccharides analysis.

The terminal tetra-saccharide GalNAcβ1-3(Fucα1-2)Galβ1-3GlcNAc could be further substituted by a Fuc in the α1-4 position of the GlcNAc residue, typifying *N*-glycans elongated with one (CG61, 70, 72, and 77) or two Le^b^/A-group motifs (CG87 and 88), as established by the presence of a B-ion at *m*/*z* 1079 and a signal at *m*/*z* 678, which was attributed to the GalNAc(Fuc)Gal C-ion (Figure 4H).

#### 2.3.2. *N*-Glycome of *O. edulis*

Using MS and MS/MS, a similar analytical pipeline applied to *O. edulis* (Figure 5) led to the identification of 59 different *N*-glycans isolated from the gills, digestive tract and mantle, versus 88 in *C. gigas*. The decreased number of *N*-glycans in *O. edulis* compared to *C. gigas* was mainly due to the lower occurrence of xylose-substituted *N*-glycans on internal Man (11 versus 37 *N*-glycans in *C. gigas*). Even when identified, the corresponding *N*-glycans detected using MS were very low-intensity signals, whereas these were observed as major signals in *C. gigas* (Figure 3A,B). The lower occurrence of Xyl substitution was further confirmed using GC/MS composition analysis, as shown later in the present study. Conversely, the occurrence of *N*-glycans with Hex core substitution was higher in *O. edulis* compared to *C. gigas* (16 vs. 9).

As for *C. gigas*, the *N*-glycans were distributed in paucimannose (24), oligomannose (7), hybrid (1) and complex families (27) (Appendix A), according to the substitution pattern in the terminal Man residues of the Man_3_GlcNAc_2_ penta-saccharide core with a combination of single GlcNAc and LacNAc-based complex antennae. According to the MS/MS fragmentation analyses, the vast majority of LacNAc-substituted *N*-glycans comprised one and two antennae, except for a single tri-antennary complex *N*-glycan (OE59) observed at *m*/*z* 3952. Three *N*-glycans (OE40, 55 and 57) were substituted by the LacNAc_2_ repeat, but the vast majority of LacNAc motifs were substituted by both Fuc and GalNAc that define the A-group antigen (OE33, 39, 41, 44, 45, 47–49, 51–59) and for some of them, an additional Fuc on the GlcNAc that defines an ALe^b^ antigen (OE44, 49 56–58). In contrast to *C.gigas*, seven *N*-glycans (OE29, 31, 34–36, 43, 52) were characterized by the presence of a terminal Fucα1_2Galβ1-3GlcNAc motif, owing to an intense Z-ion at M+Na-410, resulting from the fragmentation of the Galβ1-3GlcNAc linkage, coupled to B-fragment ion of Fucα1,2Gal1-3GlcNAc at *m*/*z* 660, as exemplified for OE35 (Figure 4J), which strongly suggests the presence of terminal type-1 H-antigen in those *N*-glycans. In OE34, the observed B-ion at *m*/*z* 834 rather than 660 suggests the presence of a Le^b^ antigen. In addition to the presence of H and Le^b^ Ag, the MS^2^ analysis of *N*-glycans OE37 and OE42 revealed the presence of a terminal disaccharide GalGal from the C/Z cleavage, which occurred between Gal and GlcNAc in the LacNAc antennae. The C/Z cleavage also confirmed the existence of the GalGal sequence at the terminal position. Moreover, the B-ion at *m*/*z* 864 revealed the presence of a Fuc residue on the GlcNAc of the antennae (Figure 4K for OE 42). The comparative analysis between organs established that all three tissues from *O.edulis*, i.e., the gills, digestive tract and mantle, expressed essentially similar *N*-glycan profiles, although the digestive tract expressed a slightly lower structural diversity (Table 1).

Altogether, the comparison of the *N*-glycomes from both species revealed not only a large variability in terms of diversity, where *C. gigas N*-glycome is much more diverse than *O. edulis*, but most importantly, in terms of antigenic presentation. Out of the 109 different *N*-glycans identified using mass spectrometry analysis of permethylated derivatives, only 38 were commonly expressed in both species (Figure 6A). However, most of these common *N*-glycans were part of oligomannosylated and paucimannosylated families. In contrast, *N*-glycans substituted by complex branches exhibited very low similarities between species, with only 13 identified common *N*-glycans out of 65 (Figure 6A). (Le^b^)/A-group appeared as the only common antigenic hallmarks in both species, which otherwise differed widely in the expression of terminal epitopes: type-1 and type-2 LacNAc for *C. gigas* and Gal-Gal and (Le^b^)/H-group for *O. edulis* (Figure 6B).

### 2.4. Methyl N-Glycome Analysis

#### 2.4.1. Localization of Methyl Groups on HBGAs

Based on the sequence and composition analyses, the methylated monosaccharides Me-3/4*O*-Fuc, Me-3/4*O*-Man and Me-3*O*-GalNAc were further localized on the *N*-glycans isolated from the different samples using mass spectrometry following per-deuteromethylation, with the help of the 3 m.u. differences it generates compared to its predicted per-deuteromethylated form. Consequently, mono-, di-, tri- and tetramethylated *N*-glycans isomers were differentiated based on the observation of -3, -6, -9 and -12 m.u. deviations from MS analyses. The positions of methylated monosaccharides within individual glycans were established with MS^2^ and MS^3^ fragmentation (Appendix A). In total, MS analyses of per-deuteromethylated *N*-glycans allowed the identification of 104 compounds substituted by 0 to 4 methyl groups in *C. gigas* and 49 in *O. edulis*, all derived from the basic structures identified after permethylation. It should be noted that all the *N*-glycans identified as permethylated derivatives could not be retrieved as per-deuterometylated derivatives due to lower sensitivity, resulting either from a lower efficiency of the substitution by I-CD_3_ and/or the isomeric dispersion for individual *N*-glycans caused by multiple methylation states. Per-deuteromethylated *N*-glycans were further analyzed using MS^2^ and MS^3^ in order to localize individual methyl groups on previously identified epitopes. Irrespective of the origin, Me-3/4*O*-Man residues were systematically localized on the Man_3_GlcNAc_2_ core of *N*-glycans from oligomannosylated, paucimannosylated and (pseudo)hybrid families, owing to intense Y- and Y/B-fragmentation ions generated by the cleavage of GlcNAc linkages. The values depended on its substitution by Xyl and Fuc residues, as exemplified by the observation of a Y-ion at *m*/*z* 1203 and a B/Y secondary ion at *m*/*z* 914 (Figure 7A) or a Y-ion at *m*/*z* 1549 and a B/Y ion at *m*/*z* 1079 (Figure 7B). In accordance with the literature that reports the presence of methylated mannose on the Man3- and/or Man6 of the trimannosylated core in numerous invertebrates, we postulated that the latter methyl group was localized on the terminal Man residue [42].

Then, the Me-3*O*-GalNAc and Me-3/4*O*-Fuc residues could be localized on the different HBGA-like terminal epitopes, as identified above in the two species of oysters (Figure 7A). In *C. gigas*, as exemplified using the MS^2^ analysis of the dimethylated *N*-glycan CG48 at *m*/*z* 2118, one methyl group was localized on the terminal A-group epitope, owing to the intense C/Z couple ions at *m*/*z* 702/1439, whereas another one was localized on the trimannosylated (Figure 7A). The MS^3^ analysis of the C-ions at *m*/*z* 702 demonstrated that the methyl group on the A-antigen was exclusively substituting the terminal GalNAc monosaccharide, owing to the observed C/Z couple ions at *m*/*z* 291/434, and in agreement with the absence of the CH_3_- group on the Fuc residue, as demonstrated by the Y-ion at *m*/*z* 505 and X-ions at *m*/*z* 280 and 310 (Figure 7A). A similar analysis of the trimethylated isomer of CG70 at *m*/*z* 2642 showed the presence of two methyl groups on the terminal Le^b^ A-group pentasaccharide, owing to the B/Y couple ions at *m*/*z* 1116/1549, out of which one was substituting the terminal A-group trisaccharide, as shown by the intense C/Z couple ions at *m*/*z* 702/1962 (Figure 7B). The MS^3^ analysis of the C-ions at *m*/*z* 702 also demonstrated the presence of a terminal Me-*O*-GalNAc residue, as in CG48. Considering that the composition analysis did not show the presence of Me-*O*-GlcNAc, the second methyl group on the antenna was localized on the Fucα1-4 residue of the Le^b^ epitope. Coupling the monosaccharide composition analysis (shown in Figure 2) with the systematic MS^2^ and MS^3^ analyses on per-deuteromethylated *N*-glycans indicated that the HBGAs of complex *N*-glycans isolated from *C. gigas* were partially methylated on the C3 position of the terminal GalNAc residue of the A-group antigen and on the C3 position Fucα1-4 residue of the Le^b^ antigen. This generated a mixture of unmethylated, mono- and dimethylated A- and ALe^b^-antigens as depicted on Figure 6C.

A similar analytical approach used on *N*-glycans from *O. edulis* showed that complex *N*-glycans from this species could be similarly substituted by a methyl group on the terminal GalNAc residue of A-group Ag as well on the Fuc residue of the disaccharide Fucα1-4GlcNAc of the A-Le^b^ epitope. In addition, the MS^2^ and MS^3^ analyses of multimethylated complex *N*-glycans showed that the Fucα1,2 that substitutes the Gal residue of the A antigen could also be methylated, contrarily to *C. gigas*, where it is not. As exemplified by the fragmentation of dimethylated OE39 at *m*/*z* 2296, the MS^2^ C/Z couple ions at *m*/*z* 699/1619 demonstrated the presence of two CH_3_- groups on the terminal GalNAc_1_Fuc_1_Gal_1_ trisaccharide that were localized on the terminal GalNAc and H-type Fuc residues, using MS^3^ analysis of C ion at *m*/*z* 699, owing to Y-ions at *m*/*z* 505 and 431 (Figure 8A). The GalNAc residue and the two Fuc residues could also be simultaneously methylated to form a trimethylated ALe^b^ epitope, as observed for the tetramethylated OE44 (Figure 8C). Finally, the MS^2^ fragmentation analysis of the H-antigen-containing *N*-glycans confirmed that the terminal Fucα1,2 residue of the H-disaccharide could also be methylated, owing to the observation of the B/Y couple ions at *m*/*z* 685/1383 and the Z-ion at *m*/*z* 1619 (Figure 8B). Altogether, the analyses showed that *N*-glycans from *O. edulis* were substituted by a combination of A- and ALe^b^-antigens methylated on the C3 position of the terminal GalNAc residue and on the C4 positions of the H- and Lewis Fuc residues, as well as on the C4 positions of the Fuc residue of the terminal H-antigen, as depicted in Figure 6C.

#### 2.4.2. Relative Quantification of Methylated Antigens

To evaluate the proportion of differentially methylated A and ALe^b^ antigens in the gills, mantle, and digestive tract of *C.gigas* and *O.edulis*, the deuteromethylated *N*-glycans were further subjected to comprehensive data-dependent HCD-MS^2^ acquisition using reverse phase C18 nano-LC-MS^2^ analysis [43]. Under the acidic solvent conditions used, the parent glycans were mostly protonated, which favored primary MS^2^ cleavages at HexNAc to produce nonreducing terminal oxonium ions, often accompanied by the loss of the terminal residue with secondary glycosidic cleavages. Each of the distinctive MS^2^ fragment ions produced and measured at high mass accuracy (<5 ppm) would identify a specific terminal glycotope or its substructure. The summed intensity for each of these target ions was extracted from all the MS^2^ spectra acquired across the entire LC range, where the glycans eluted are indicative of their respective glycomic abundance in comparative analyses. This allowed the relative abundance of the A and ALe^b^ antigens, differing only in their methylation status and distributed over a diverse range of parent *N*-glycans, to be relatively quantified. As shown in Figure 9, the glycans carrying the differentially methylated A antigen produced the characteristic MS^2^ ions at *m*/*z* 919 (no CH_3_), 916 (1 CH_3_) and 913 (2 CH_3_). The additional loss of Fuc produced the ions at *m*/*z* 722 (no CH_3_) and 719 (1 CH_3_), whereas the loss of the terminal GalNAc yielded the ions at *m*/*z* 648 (no CH_3_) and 645 (1 CH_3_). The overall pattern was consistent with both the terminal GalNAc, and Fuc of A antigen could be methylated but more completely than for the GalNAc. Similarly, glycans carrying the ALe^b^ antigen with up to three methylations produced the ions at *m*/*z* 1099 (no CH_3_), 1096 (1 CH_3_), 1093 (2 CH_3_) and 1090 (3 CH_3_), while further loss of the mostly methylated terminal GalNAc produced the ions at *m*/*z* 828 (no CH_3_), 825 (1 CH_3_) and 822 (2 CH_3_). The methylation status of the extra Fuc on the GlcNAc of ALe^b^ could be relatively quantified using the ions at *m*/*z* 417 (no CH_3_) and 414 (1 CH_3_) for the internal Fucα1,4GlcNAc disaccharide moiety.

The identification and relative quantification of each of the target MS^2^ ions in the PNGase F-released *N*-glycans confirmed that the A antigen from both *C. gigas* and *O. edulis* was almost completely methylated at the terminal GalNAc, whereas the Fucα1,2 was methylated at a significant level only in *O. edulis*. This result was shown by the detection of dimethylated A antigen (*m*/*z* 913) and monomethylated internal Fucα1,2Gal1,3βGlcNAc (*m*/*z* 645) fragments only in *O. edulis*. The relative quantification of these ions further indicated that the A antigen was almost completely monomethylated in the three organs of *C. gigas*, although with less than 10%of the unmethylated form in the digestive tissue (Figure 9A). The analysis of *O. edulis* tissues, on the other hand, showed that the terminal A antigen in all three tissues was more than 80% substituted by at least one methyl group, whereas a significant proportion was dimethylated (Figure 9B). The MS^2^ ions at *m*/*z* 722/719 and *m*/*z* 648/645 further indicated that more methylated Fucα1,2 was found in the mantle of *O. edulis* than in the other two tissues, consistent with a much higher proportion of dimethylated A antigen in this tissue (>50%) compared to the other two (<25%) (Figure 9B). For the ALe^b^ antigen, the ion intensity ratio of signals at *m*/*z* 414 vs. 417 showed that the extra Fucα1,4 residues could be methylated in both species, with much higher levels in the gills and mantle relative to digestive tissue. Due to this extra methyl, the ALe^b^ antigen can carry up to three methylations, particularly in *O. edulis*, since it also has a significantly higher level of methylation on the other Fuc compared to *C. gigas*. However, due to the preferential MS^2^ cleavage of the terminal GalNAc residue, the overall signal intensities of the signals associated with the methylated forms of the ALe^b^ antigen were very low and could not be reliably quantified.

### 2.5. Molecular Modeling of Norovirus P Domain Interaction with Methylated A Antigens

In order to determine the potential influence of methyl groups decorating the terminal A antigen trisaccharide on the interactions between oyster saccharides and human noroviruses from different genogroups, the interaction maps of a GI.1 (Norwalk virus) and GII.3 (TV24) with non-methylated Human A-antigen were compared with those of mono- and dimethylated epitopes identified in *C. gigas* and *O. edulis*. The X-ray crystallographic structures in the P domain of GI.1 (PDB code 3D26) and GII.3 (PDB code 6IS5) viruses in complex with A-trisaccharide or A-tetrasaccharide, respectively, were used as patterns. The glucose in the A-tetrasaccharide was removed and hydroxyls at position 3 of GalNAc and 4 of Fuc in the trisaccharides were substituted with a methyl group to model saccharides exposed on the surface of the oyster species *Crassostrea gigas* and *Ostrea edulis*. Non-bond interactions between the atoms of viruses and those of saccharides were monitored using BIOVIA Discovery Studio visualizer (Dassault Systèmes, San Diego). First, the recognition between GI-1 and oyster epitopes appeared to differ in the loss of two hydrogen bonds (HB) between the oxygen (O) at position 3 of GalNAc and the residues Asp327 and Ser377 of GI.1 P domain, due to the methylation of this position in both *C. gigas* and *O. edulis*, thus potentially weakening the interaction (Figure 10).

However, the GI.1 strain seems to recognize both saccharides Me-3*O*-GalNAcα1-3(Fucα1-2)Galβ and Me-3*O*-GalNAcα1,3(Me-4*O*-Fucα1,2)Galβ in a similar manner. The oxygen (O) at position 4 of GalNAc makes a hydrogen bond (HB) with Pro378, whereas the O2 of Fuc makes an HB with Ser380 of GI-1 (Figure 10 A). In comparison, recognition by GII.3 seems stronger as it involves a larger number of HBs (Figure 10 B). The Fuc residue makes five HB with Gly451, Asp386, Thr357 and Arg358 and two carbon HBs (CHBs) with the Thr356 and Thr357 of TV24 virus, and the GalNAc residue makes one HB with Lys363 of TV24 when only GalNAc is methylated at position 3. The methylation of Fuc at position 2 causes the loss of two HBs and one CHB. These observations suggest that the attachment of the GII.3 strain to the glycans in the *C. gigas* oyster species is stronger than that in the GI.1 strain. However, the attachment of the latter strain should be similar between the two oyster species, whereas binding of the GII.3 virus by *Ostrea edulis* may certainly be weaker than that *by Crassostrea gigas* depending on the quantity and proportion of Me-3*O*-GalNAcα1,3(Fucα1,2)Galβ and Me-3*O*-GalNAcα1,3(Me-4*O*-Fucα1,2)Galβ in *O. edulis*.

## 3. Discussion

Although belonging to the same Ostreoidea superfamily, the two oyster species *Crassostrea gigas* and *Ostrea edulis* showed very different *N*-glycan profiles in terms of variability, modification to the ManGlcNAc_2_ core, expression of terminal antigens and methylation patterns. Of the 109 unique *N*-glycans identified, i.e., 88 in *C. gigas* and 59 in *O. edulis,* excluding the variability generated by methylation modification, only 38 were common to both species (Figure 6). Most of these common glycans are oligomannosylated and paucimannosylated *N*-glycans, leaving only 13 that present complex-type branches commonly expressed in both species.

These *N*-glycans exhibit many structural features of the Man_3_GlcNAc_2_ core that have been previously described in invertebrates [29]. Among those, the complex-type *N*-glycans showed a high proportion of core fucosylation, either in α1-6 linkage or α1-3 linkage, out of which some simultaneously present both α1-6 and α1-3 fucose residues (about 12% in *C gigas* and 20% in *O. edulis*) that are already described in invertebrates [44]. Fucα1-3 has been widely detected outside human glycoproteins in plants [45], insects [46] and nematode [47] and has been shown to be transferred by α-1-3 fucosyltransferase (FUT-1) [48]. Based on data collected from *Caenorhabditis elegans* and *Drosophila melanogaster*, it was suggested that α1,3 and α1,6 core fucoses are sequentially added to the chitobiose core by FUT1 and α-1,6-fucosyltransferase (FUT-6) [48,49]. Among invertebrate-specific core modifications, we also observed the presence of a disaccharide Fuc-Hex on the reducing terminal GlcNAc residue. Here, MALDI-MS/MS did not allow us to discriminate between Fuc-Hex or Hex-Fuc disaccharides or to identify the nature of hexose as in Gal-Fuc motif, as previously reported in results from the galactosylation of the core α1-6Fuc by GALT-1 galactosyltransferase [29,50]. This unusual galactosylation has also been reported for α1-3Fuc linked to either the reducing terminal GlcNAc and/or the non-reducing terminal GlcNAc in the core [47]. We also observed the presence of a single Hex residue on the terminal-reducing GlcNAc of numerous *N*-glycans in both species (26% of *N*-glycans in *O. edulis* and 11% in *C. gigas*). A similar modification was observed in *Volvarina rubella*, a margin snail of the clade Neogastropoda [51], and in the oyster *C. Virginica* [30], which was described as a Man residue based on its sensitivity to β-mannosidase [30,51]. When present, this Hex was observed either alone or in combination with a single Fuc residue or, more rarely, associated with a second Hex on the terminal reducing GlcNAc in *O. edulis*. The nature of this core modification was not investigated further because the main objective of the present study was the terminal complex epitopes that may serve as ligands for viruses. Finally, xylosylation of internal Man residue was shown to be a common feature of *N*-glycans isolated from *C. gigas* and *O. edulis*, although in varying amounts in each species (about 42% of *N*-glycans are substituted by Xyl residues in *C. gigas* and 19% in *O. edulis*), as observed in numerous species of parasites and gastropods [52,53,54]. Noticeably, no Xyl residue was found on *N*-glycans isolated from the pacific oyster *Crassostrea virginica*, which strongly suggests that this modification is highly species-specific [30].

More than half of the complex-type *N*-glycans (51% in *C. gigas* and 62% in *O. edulis*) were substituted by a terminal A-antigen motif, associated or not with an internal Le^b^ motif to form the ALe^b^ motif, and a minority by a terminal H-antigen in *O. edulis* (14%), but none in *C. gigas*. All GalNAc residues identified in the *N*-glycans from both *C. gigas* and *O. edulis*, irrespective of the tissue, are part of either A- or ALe^b^-Ag, suggesting that the amount of GalNAc residues is a direct reflection of the amount of terminal A-antigen present in the tissues of the two oyster species. With the exception of a single *N*-glycan, the same conclusion can be drawn for *C. virginica* [30]. The quantification of GalNAc in *N*-glycans (Appendix A) demonstrates that tissues from *O. edulis* express between two (in the digestive track) and five times (in the gills and mantle) more A- and ALe^b^-Ag than tissues from *C. gigas*, which results in increased exposure of potential glycan–ligands of noroviruses on *O. edulis* compared to *C. gigas*.

The two species also differ in their methylation patterns, as demonstrated by the detailed localization of methyl groups on monosaccharides of individual *N*-glycans in a tissue-specific manner. In both species, most of the GalNAc residues were shown to be methylated at their C3 position, although in different proportions according to the tissues: about 55% in the gill (CG 54% and OE 59%), 66% in the digestive track (67% in CG and 65% in OE) and 78% in the mantle (79% in CG and 76% in OE). In comparison, about 37% of the GalNAc was reported to be methylated in the Pacific oyster *C. virginica* [30]. About 25% of the fucose was also shown to be methylated in *C. gigas* and *O. edulis* (19% in CG and 29% in OE), whereas no methylated fucose was observed in *C. virginica* [30]. However, in contrast to GalNAc, which was methylated at the C3 position in both species, Fuc residues were methylated in a species-specific manner at the C3 and C4 positions in *C. gigas* and *O. edulis*, respectively, demonstrating the existence of an exquisitely specific methylation process. It should be noted that both 3-*O*-Me-GalNAc and 3-*O*-Me-Fuc were previously identified in glycolipids isolated from the freshwater mussel *Hyriopsis schlegelii*, but to our knowledge, the presence of 4-*O*-Me-Fuc has never been reported in invertebrates [55]. Similar to GalNAc and Fuc, about 25% of the Man residues in *N*-glycans isolated from the three tissues of the two species were methylated. However, 3-*O*-Me-Man and 4-*O*-Me-Man were simultaneously identified in both species but in very different ratios according to the tissues (Figure 2).

Surprisingly, our analyses did not reveal any sulfation on the *N*-glycans in *C. gigas*, in contrast to the Pacific oyster, the glycans of which were shown to be heavily sulfated on galactose residues, either in the terminal position or within the A antigenic motif [30]. Sulfation is indeed a ubiquitous and well-known modification affecting various monosaccharides, such as the galactose or glucuronic acid residues in humans or sialic acid in sea urchins [56,57]. However, sulfated *N*-glycans are notoriously difficult to detect using mass spectrometry-based glycomic analysis due to ion suppression, which occurs regardless of the positive or negative mode chosen. Furthermore, during the classical methylation protocol, permethylated sulfated *N*-glycans are totally or partially lost during the chloroform/water cleaning procedure due to their solubility in water. To overcome this problem, solid phase extraction using a weak anion exchange cartridge was used to recover putative sulfated *N*-glycans, as previously demonstrated [58,59]. Several attempts using this optimized procedure coupled to MALDI-MS analysis operating in negative and positive modes failed to identify any sulfated compound, strongly suggesting that *N*-glycans of *C. gigas* are devoid of sulfate substitution, in contrast to *C. virginica*.

Although the two oyster species studied here present HBGA motifs, these were shown to be, for a large part, modified by methyl groups in contrast with human HBGAs. These modifications may affect the binding and bioaccumulation of norovirus strains. To begin exploring this possibility, we modeled the interaction between two viral strains and different versions of the A-group epitope. The Norwalk and TV24 strains were used for the analysis because their binding modes to HBGAs are representative of the GI and GII binding modes in the majority of circulating human strains [60,61,62]. The modeling results should be considered with caution since they do not take into account dynamic aspects of the virus–glycan interactions or the effect of glycans clustering. They nonetheless show relevant potential differences between the GI and GII attachment to the two oyster species. Because the GII.3 strain interaction with *C. gigas* involves more HB than that of the GI.1 strain, it suggests a higher binding of GII.3 strains to tissues of that oyster species. This might explain the previously reported binding and bioaccumulation of GII to a larger set of oyster tissues [19,25,26,28,63]. Interestingly, the interaction for the GI strain does not appear to be different between the two oyster species *C. gigas* and *O. edulis*, whereas that of the GII strain is expected to be much lower in the latter species, according to the amount and tissue distribution of methylated α1,2-linked fucose. These observations suggests that the attachment of norovirus to oyster tissues may not only be virus strain-dependent but also dependent on species, *O. edulis* being possibly less prone to bioaccumulate epidemiologically dominant viral strains (personal observation). These hypotheses should be experimentally demonstrated, ideally using saturation transfer difference (STD) NMR spectroscopy, which provides a direct mapping of the glycan epitope recognized with VLP. This approach allowed us to precisely map the glycan epitopes recognized by norovirus epitopes out of synthetic type 1, type 2, type 3, type 5, and type 6 blood group A- and B-tetrasaccharides without relying on X-ray crystallography analysis of carbohydrate–protein complexes, which could be directly applied to differentially methylated A-antigens either from a natural source or generated with chemical synthesis [64,65].

## 4. Materials and Methods

### 4.1. Preparation of Anatomic Pieces

*Crassostrea gigas* and *Ostrea edulis* adult oysters of commercial size were bought live in November 2018 from a local shellfish producer (Nantes, France). Upon arrival at the laboratory, oysters were shucked, and then their body was removed from the shell and dissected on ice using sterile scalpels and pliers. Three tissues (the gills, the mantle, and the digestive tract) were collected from 10 to 15 individuals, pooled per tissue type, and stored at −80 °C Before use, tissues were homogenized for 2 min in water using gentleMACS™ Dissociator (Miltenyi Biotec, Bergisch Gladbbach, Germany), freeze-dried and stored at −20 °C until further use.

### 4.2. Preparation of N-Glycans

The digestive tract, gills or mantle (100 mg) were suspended in 1 mL of PBS containing 1% of triton X-100 and sonicated for 2 min in an ice bath. The mixture was stirred at 37 °C overnight. After centrifugation at 13,000 rpm for 10 min at room temperature, proteins and glycoproteins contained in supernatants were reduced with 100 μL of a 0.1 M DTT in PBS solution at 37 °C for 1 h under stirring and then alkylated by adding 110 μL of a 0.5 M iodoacetamide in PBS solution and incubated at 37 °C for 1 h under stirring. Proteins and glycoproteins were precipitated by adding 130 μL of a saturated trichloroacetic acid solution and placed at −20 °C for 30 min. After centrifugation at 13,000 rpm for 10 min at 4 °C, the pellet was washed 3 times using 1 mL of cold acetone (−20 °C). After drying with a vacuum Speed Vac (Eppendorf, Hamburg, Germany) (5 min.), 500 μL of a 50 mM NH_4_HCO_3_ solution was added, followed by 40 μL of a 2 mgmL^−1^ trypsin in a 50 mM NH_4_HCO_3_ solution and then incubated at 37 °C for overnight. Contaminating glycogen fragments were removed using a C18 SPE cartridge equilibrated in a 5% acetic acid solution (10 mL). Peptides and glycopeptides were recovered by successively passing 10 mL of 20%, 40% and 60% 2-propanol in 5% acetic acid. After freeze drying, they were resuspended in 500 μL of a 50 mM solution, and 1 μL of a PNGase F (62,5 IU) was added to release *N*-glycans, which were recovered in the unretained fraction of a C18 SPE cartridge operated as previously. Peptides and insensitive PNGase F glycopeptides were recovered by pooling the 3 different isopropanol fractions (20, 40 and 60%), which were freeze-dried before digestion with PNGase A according to the manufacturer’s recommendations.

### 4.3. Protein Assay

Protein concentrations were measured in triplicate using a Micro BCA™ Kit (Thermo scientific, Rockford, IL, USA) in a 96-well microplate following the manufacturer’s recommendations and using Bovine Serum Albumin 1 mg/mL as standard.

### 4.4. Permethylation of N-Glycans

*N*-glycans were dissolved in 400 μL of a slurry base extemporaneously prepared as follows: 400 μL of a 50% sodium hydroxide in water was vigorously mixed to 800 μL of methanol and then 4 mL of DMSO. After centrifugation at 1500 rpm for 1 min at room temperature, the insoluble reagent was removed, and the resulting clear solution was washed again with 4 mL of DMSO. The operation was repeated until no insoluble compound was observed. Iodomethane or deuterated iodomethane (200 μL) was added to the slurry base, and the methylation was achieved in 30 min at room temperature under stirring. The reaction was stopped by adding 2 mL of a cold 5% acetic acid solution and 1 mL of chloroform. The aqueous phase was removed, and the chloroform phase was washed 5 times with 2 mL of water before drying under a stream of nitrogen.

### 4.5. MALDI-MS Analysis

The permethylated or deuteromethylated *N*-glycans were solubilized in 100 μL of ACN, and 1 μL was mixed with 1 μL of a matrix solution prepared by dissolving 10 mg.mL^−1^ of 2,5-dihydroxybenzoic acid in 1 mL NaOH 1 mM and spotted on a MALDI plate. MS and MS^n^ spectra were acquired using a 4800 Proteomics Analyzer mass spectrometer (Applied Biosystems, Framingham, MA, USA) and using an Axima Resonance spectrometer (Shimadzu, Kyoto, Japan), respectively, in positive reflection modes. For MS^2^ experiments, collision energy was tuned from 300 to 600 eV, and argon was used as collision gas with a CID control manually adjusted (between 80∼130) to achieve an optimum degree of fragmentation. For MS^3^, ion selection occurred with a standard resolution with the same collision energy.

### 4.6. Quantification of A and ALe^b^ Epitopes Using NanoLC-MS/MS

Deuteromethylated glycan samples were analyzed with reverse phase C18 nanoLC-MS/MS using an Orbitrap Fusion Tribrid Mass Spectrometer system (ThermoFisher Scientific) fitted with a ReproSil-Pur 120 C18-AQ column (120 Å, 1.9 µm, 75 μm × 200 mm, Dr. Maisch), using the same LC conditions and data acquisition method as described previously [66]. The HCD-MS2 data acquired were processed with an in-house-developed LC-MS2/MS3 glycomic data mining tool, Glypick, as described in [43]. The intensities of the MS2 ions representing the diagnostic fragment ions of differentially methylated terminal A and ALe^b^ glycotopes that passed the preset 5 ppm filtering criteria were extracted, summed and output in Excel format. The summed ion intensities for each of the target ions, differing by 3 mass units, representing different degrees of methylation on the same deuteromethylated glycotope were calculated as % total of that structure present in each sample analyzed.

### 4.7. Preparation of Itol-Acetate Derivatives

The monosaccharides of *N*-glycans were released with 4 M TFA for 4 h at 100 °C. After drying under a nitrogen stream, traces of TFA were removed using 4 co-distillations with 1 mL of methanol. Monosaccharides were reduced for 4 h in 500 μL 50 mM ammonia in the presence of either sodium borohydride 1 M or sodium borodeuteride 1 M. Borates were removed using 9 co-distillations in methanol/acetic acid (9/1; *v/v*). Reduced monosaccharides were peracetylated in 500 μL acetic anhydride and 20 μL pyridine at 100 °C for 4 h.

### 4.8. GC-MS Analysis Itol-Acetates and Permethylated Derivatives

Itol-acetates derivatives were dissolved in 200 μL chloroform and injected in splitless mode on a Solgel 1 MS, 30 m × 0.25 mm × 0.25 μm capillary column with the following gradient temperature: 120 to 230 °C, 3 °C/min and then to 270 °C, 10 °C/min. The compounds were detected at 70 eV using a HP-7820 gas chromatograph coupled to a 5977B single quad (Agilent Technologies, Santa Clara, CA, USA) in full scan mode from 45 to 500 Da.

Permethylated monosaccharides were resuspended in 200 μL of chloroform and injected using the same GC set with the following gradient: 80 °C to 180 °C, 2 °C/min and then to 250 °C, 30 °C/min.

### 4.9. Molecular Modeling

To determine the potential interactions between oyster saccharides and human noroviruses from different genogroups, the X-ray crystallographic structures in the P domain of Norwalk (GI-1, PDB code 3D26) and TV24 (GII.3, PDB code 6IS5) viruses in complex with A-trisaccharide or A-tetrasaccharide, respectively, were used. The glucose in the A-tetrasaccharide was removed and hydroxyls at position 3 of GalNAc and 4 of Fuc and the trisaccharides were substituted with a methyl group to model saccharides exposed on the surface by oyster species *Crassostrea gigas* and *Ostrea edulis*. Non-bond interactions between the atoms of viruses and those of saccharides were monitored using the BIOVIA Discovery Studio visualizer (Dassault Systèmes, San Diego, CA, USA).

## Figures and Tables

**Figure 1 marinedrugs-21-00342-f001:**
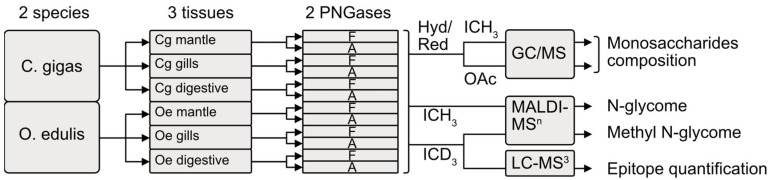
Analytical workflow used to characterise *N*-glycome in the gills, digestive tract and mantle of two oyster species, *Crassostrea gigas* and *Ostrea edulis*. Hyd/Red, hydrolysis and reduction; ICH_3_, permethylation; ICD_3_, perdeuteromethylation; OAc, peracetylation, F, PNGase F; A, PNGase A.

**Figure 2 marinedrugs-21-00342-f002:**
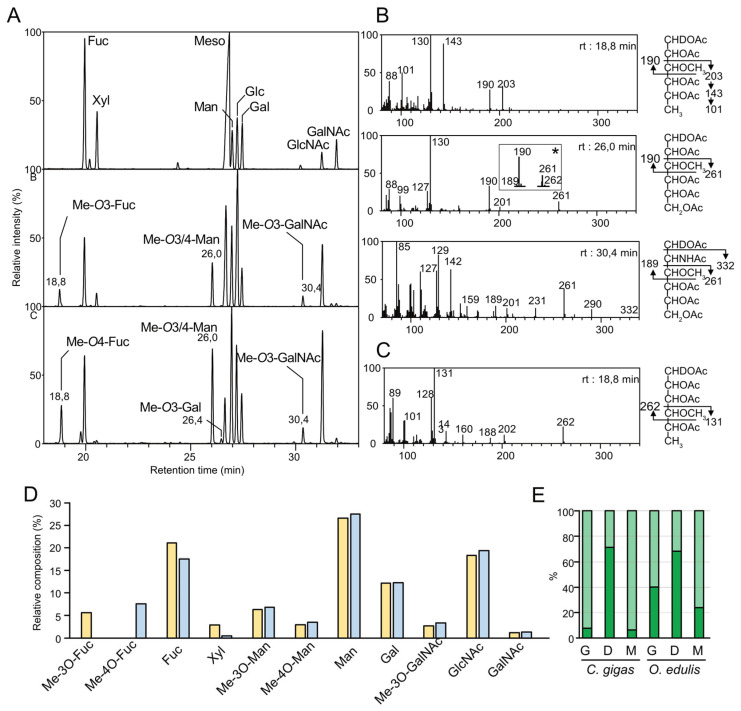
Monosaccharide composition analysis of *N*-glycans. (**A**) Total ion count generated from GC-MS analysis of itol-acetate derivatives of (top) authentic standards, (middle) *N*-glycans released from the mantle of *C. gigas* with PNGase F, (bottom) *N*-glycans released from the mantle of *O. edulis* wit PNGase F. (**B**) EI-MS fragmentation patterns of itol-acetates signals from *C. gigas* at (top) 18.8 min identified as Me-*O*3-Fuc; (middle) 26.0 min identified as Me-*O*3-Man; inset (*) corresponds to a minor signal identified as Me-*O*4-Man; and (bottom) 30.4 min identified as Me-*O*3-GalNAc. (**C**) EI-MS fragmentation patterns of the itol-acetates signal at 18.8 min from *O. edulis* identified as Me-*O*4-Fuc, all other signals were identical to *C. gigas*. (**D**) Relative quantifications (in %) of monosaccharides calculated as the sum of individual signals of *N*-glycans released with PNGase F and A from the three tissues of *C. gigas* (yellow) and *O. edulis* (blue). The percentage of Me-3*O*-Man and Me-4*O*-Man was determined by computing the ratios of signals at *m*/*z* 190/261 vs. *m*/*z* 189/262 recorded on the EI-MS spectra. (**E**) Ratios (in %) of Me-3*O*-Man (light green) and Me-4*O*-Man (dark green) in *N*-glycans isolated from individual tissues of *C. gigas* and *O. edulis* (G, gills; D, digestive tissue; M, mantle).

**Figure 3 marinedrugs-21-00342-f003:**
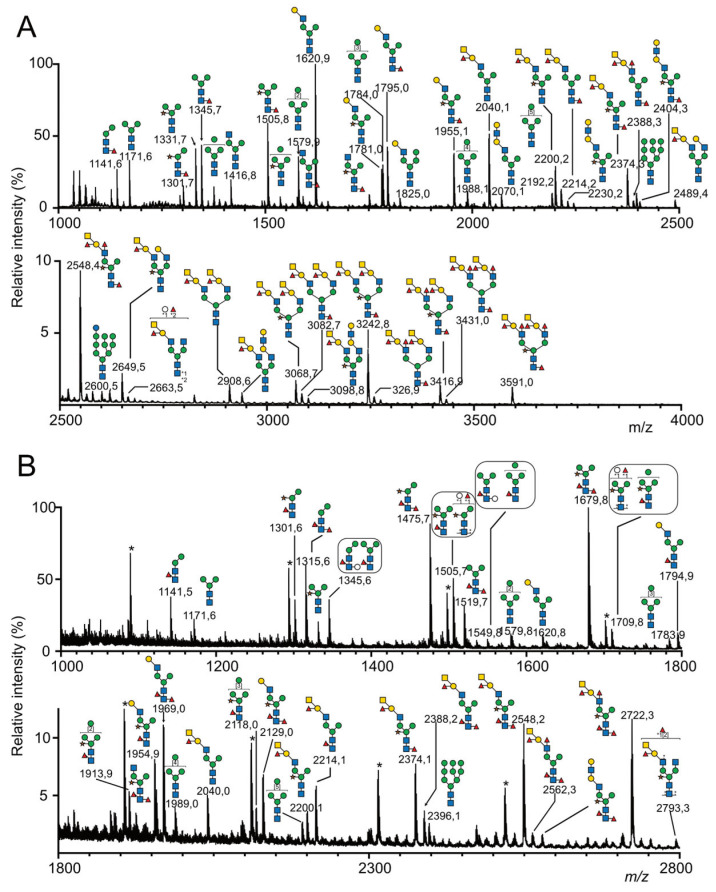
MALDI-MS analysis of permethylated *N*-glycans isolated from the mantle of *Crassostrea gigas*. Prior to their permethylation, *N*-glycans were sequentially released from mantle of *C. gigas* with (**A**) peptidyl-*N*-glycosidase F and (**B**) peptidyl-*N*-glycosidase A. Glycosyl composition assignment was based on the detected mass values for the [M+Na]^+^ molecular ions and subsequent MS^n^ sequencing. Putative *N*-glycan structures were represented according to the standard Symbol Nomenclature for Glycans system [39]. A complete list of identified *N*-glycans is provided in Appendix A. The presence of an asterisk (*) above an *m*/*z* signal indicates that no structure could be deduced.

**Figure 4 marinedrugs-21-00342-f004:**
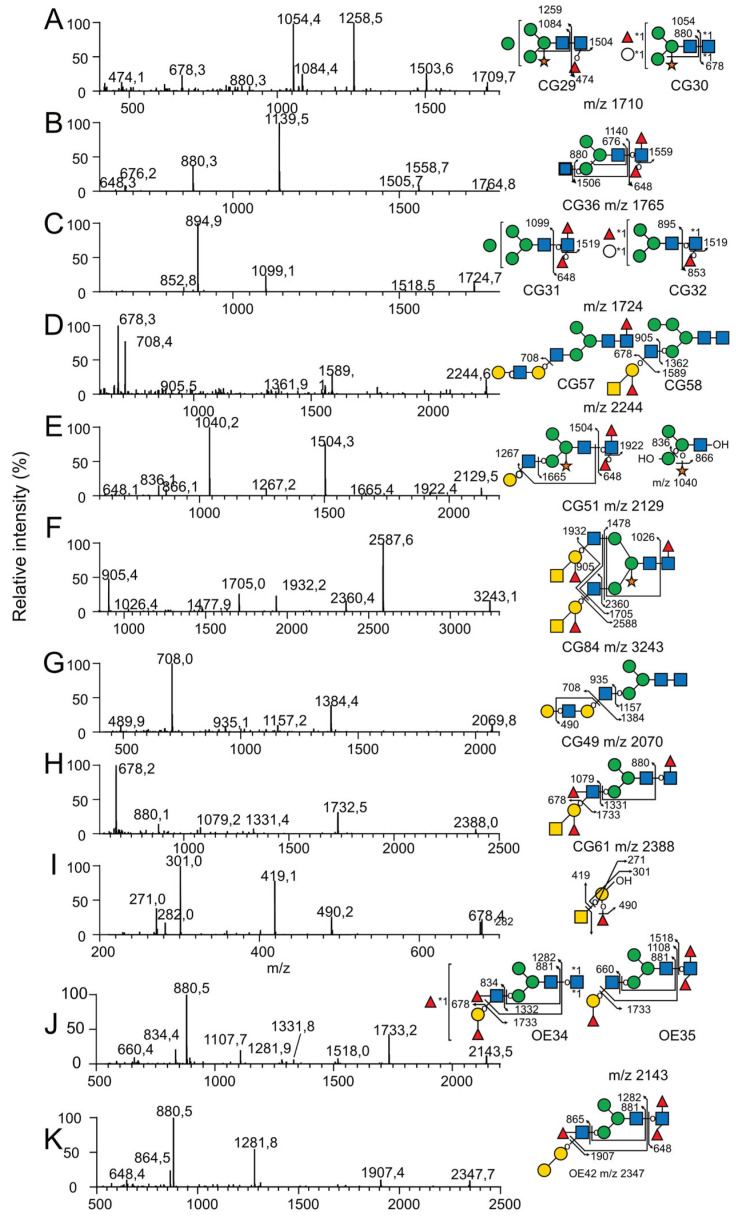
Selection of MS^2^ and MS^3^ spectra for representative permethylated *N*-glycans. MS^2^ spectra of *N*-glycans (**A**) CG29 and CG30, (**B**) CG36, (**C**) CG31 and CG32, (**D**) CG57 and CG58, (**E**) CG51, (**F**) CG84, (**G**) CG49 and (**H**) CG61 released from *C. gigas* with PNGase F. MS^2^ spectra of *N*-glycans (**J**) OE34 and OE35 and (**K**) OE42 released with PNGase A from *O. edulis.* All glycans were permethylated prior to their analysis. (**I**) MS^3^ spectrum of the *m*/*z* 678 ion generated during MS^2^ analysis of the *N*-glycan CG61. *N*-glycan structures were represented according to the standard Symbol Nomenclature for Glycans system [39].

**Figure 5 marinedrugs-21-00342-f005:**
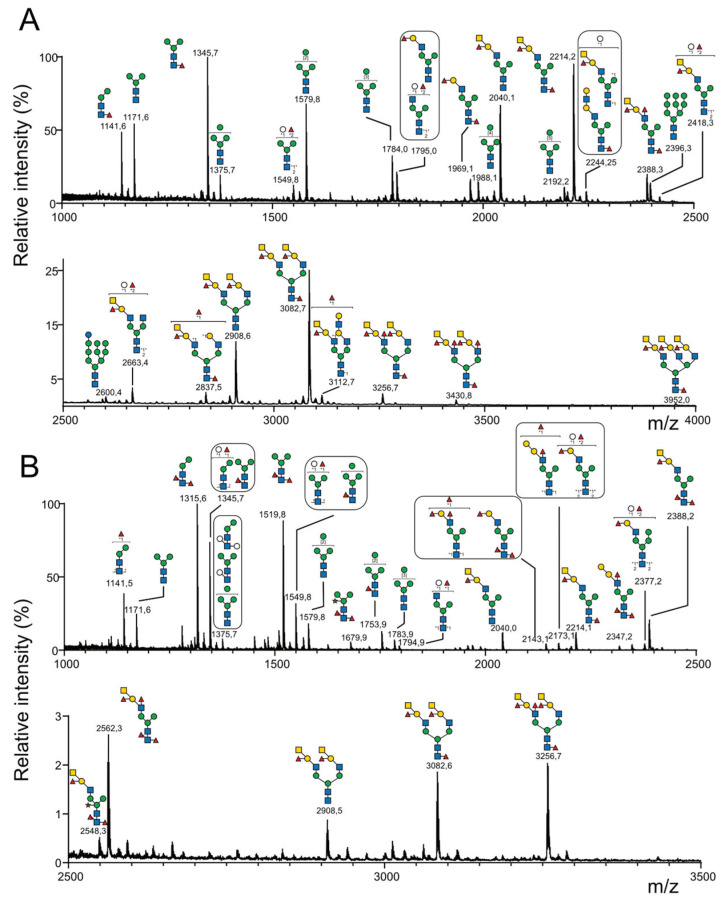
MALDI-MS analysis of permethylated *N*-glycans isolated from the mantle of *Ostrea edulis*. Prior to their permethylation, *N*-glycans were sequentially released from the mantle of *O. edulis* with (**A**) peptidyl-*N*-glycosidase F and (**B**) peptidyl-*N*-glycosidase A. Glycosyl composition assignment was based on the detected mass values for the [M+Na]^+^ molecular ions and subsequent MS^n^ sequencing. Putative *N*-glycan structures were represented according to the standard Symbol Nomenclature for Glycans system [39]. A complete list of identified *N*-glycans is provided in Appendix A. The presence of an asterisk (*) above an *m*/*z* signal indicates that no structure could be deduced.

**Figure 6 marinedrugs-21-00342-f006:**
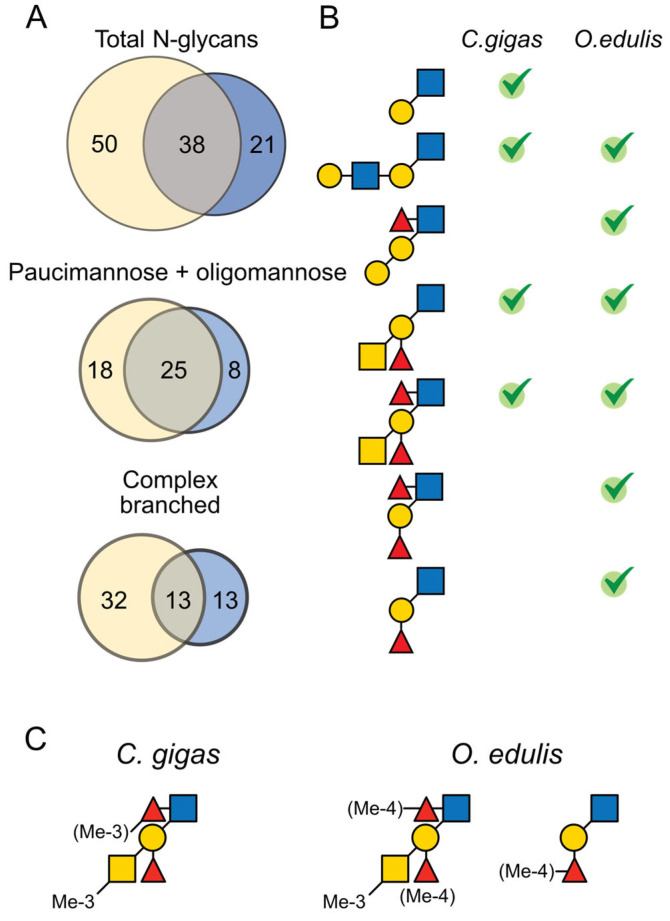
Distribution of common and specific terminal glycan epitopes in *C. gigas* and *O. edulis*. (**A**) Venn diagram showing the distribution of common epitopes in *C. gigas* (in yellow) vs. *O. edulis* (in blue). The numbers indicate the number of glycans identified in each species and the sizes of the circles and intersections are proportional to the number of *N*-glycans. The diagrams were generated using BioVenn [41]. The top diagram gathers all identified *N*-glycans; the other diagrams correspond to individual families of *N*-glycans, as depicted. (**B**) Distribution of the different oligosaccharide motifs (LacNAc, LacNAc repeat, GalGal[Fuc]GlcNAc-, A Ag, ALe^b^ Ag, Le^b^ and H-type 1 Ag from top to bottom) identified on the pentasaccharide core in the two oyster species. Ticks indicate the presence of the motif in each species. (**C**) Summary of the methylation status of terminal HBGA in *C. gigas* and *O. edulis*.

**Figure 7 marinedrugs-21-00342-f007:**
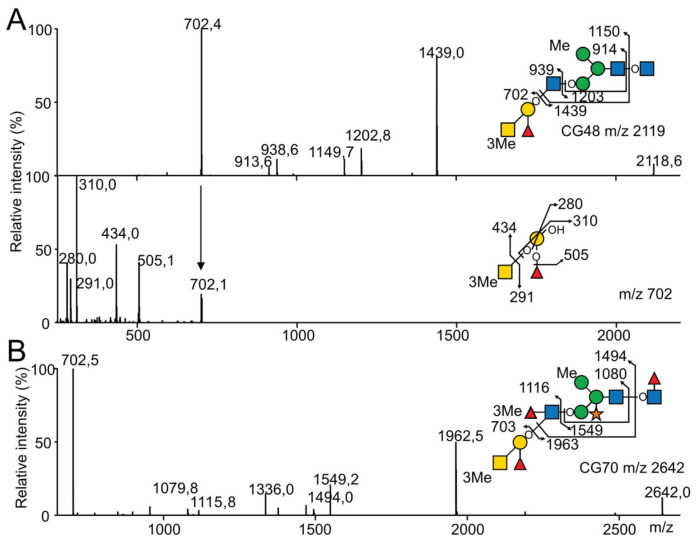
MALDI MS sequencing of the partially methylated and deuteromethylated *N*-glycans from *C. gigas* for the precise location of the methyl group. (**A**) MALDI-MS^2^ and MS^3^ spectrum of *N*-glycan CG48 and (**B**) MALDI-MS^2^ of *N*-glycan CG70. *N*-glycans were released with peptidyl-*N*-glycosidase F from *C. gigas* and permethylated with I-CD_3_ prior to their analysis. For CG48, the MS^2^ signal at *m*/*z* 702 corresponding to the monomethylated terminal A antigen was selected for MS^3^ fragmentation.

**Figure 8 marinedrugs-21-00342-f008:**
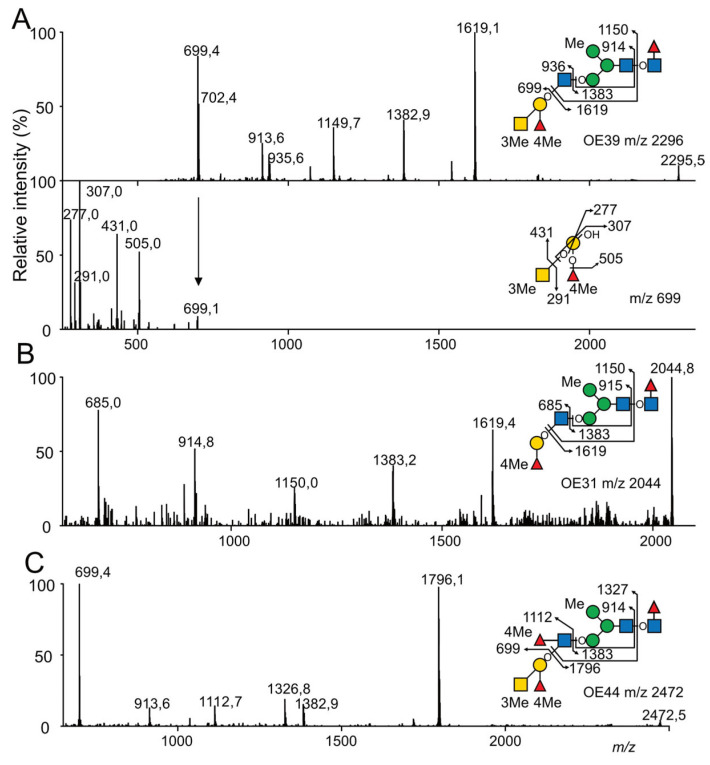
MALDI MS sequencing of the partially methylated and deuteromethylated *N*-glycans from *O. edulis* for the precise location of the methyl group. (**A**) MALDI-MS^2^ and MS^3^ spectrum of *N*-glycan OE39. MALDI-MS^2^ of (**B**) *N*-glycan OE31 and (**C**) OE44. *N*-glycans were released with peptidyl-*N*-glycosidase F from *O. edulis* and permethylated with I-CD_3_ prior to their analysis. For OE39, the MS^2^ signal at *m*/*z* 699 corresponding to dimethylated terminal A-Ag was selected for MS^3^ fragmentation.

**Figure 9 marinedrugs-21-00342-f009:**
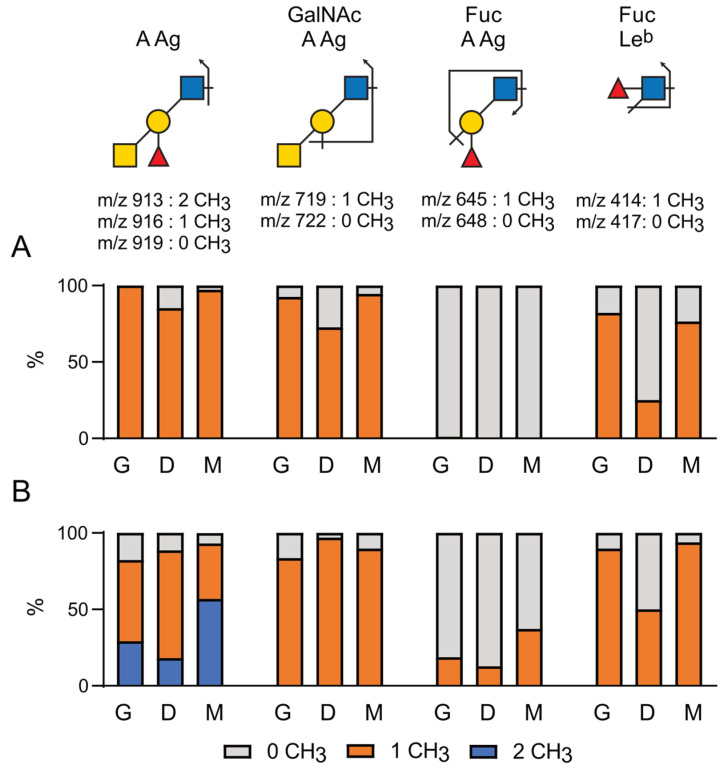
Relative quantification of the native and methylated forms of A Ag and Le^b^ Ag. Deuteromethylated *N*-glycans of *C. gigas* and *O. edulis* were analyzed using LC-MS^2^. Characteristics of the MS^2^ ions for individual antigens and their respective methylated forms were identified as depicted on top. Proportions of methylated individual glycan epitopes were quantified in *N*-glycans isolated from the three tissues of (**A**) *C. gigas* and (**B**) *O. edulis*. Light gray, 0 CH_3_ group; orange, 1 CH_3_ group; blue, 2 CH_3_ groups. G, gills; D, digestive tract; M, mantle.

**Figure 10 marinedrugs-21-00342-f010:**
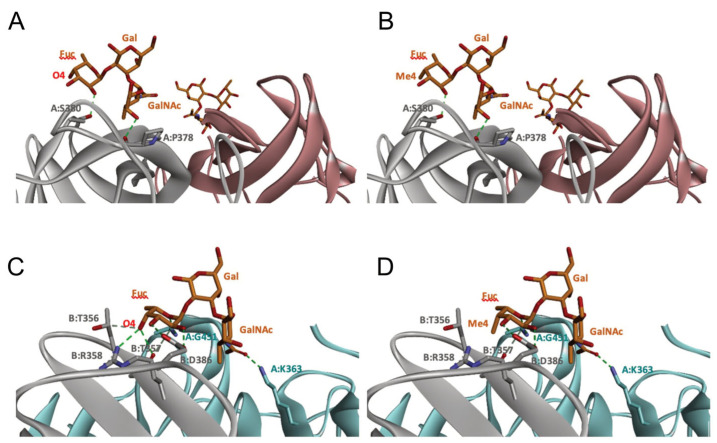
Structural models for GI.1 (Norwalk) and GII.3 (TV24) P dimers in complex with oyster trisaccharides. Structure of Norwalk (**A**,**B**) and TV24 (**C**,**D**) P dimer in complex with Me3GalNAcα1,3(Fucα1,2)Galβ (**A**,**C**) and Me3GalNAcα1,3(Me4Fucα1,2)Galβ (**B**,**D**). HB and CHB between atoms of the virus P domain and atoms of saccharides are shown with green and pale green dashed lines, respectively.

**Table 1 marinedrugs-21-00342-t001:** A 2D diagram showing the detailed HB (green dashed lines) and CHB (pale green dashed lines) network between amino acids in the P domain of viruses and individual saccharides in the A-trisaccharide.

Virus	Trisaccharides		
	α-l-Fuc-(1→2)-[α-d-GalNAc-(1→3)]-β-D-Gal	α-l-Fuc-(1→2)-[α-d-Me3GalNAc-(1→3)]-β-D-Gal	α-l-Me4-Fuc-(1→2)-[α-d-Me3GalNAc-(1→3)]-β-D-Gal
**Norwalk**	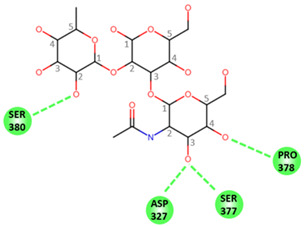	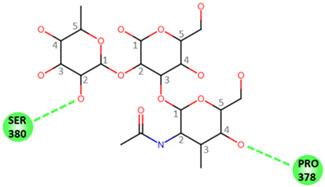	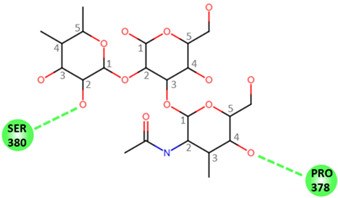
**TV24**	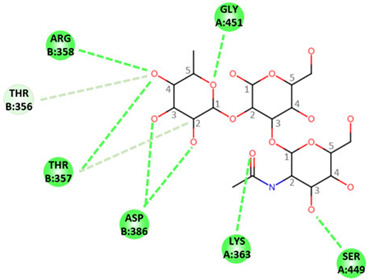	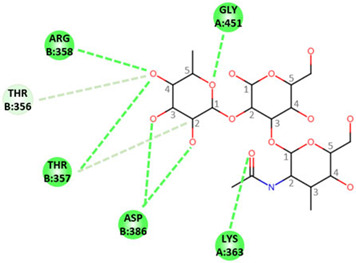	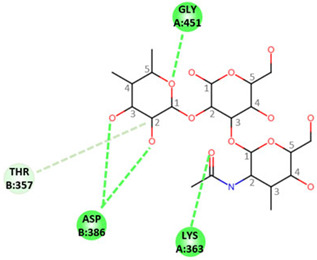

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
