# Peer review of "Species-Specific N-Glycomes and Methylation Patterns of Oysters Crassostrea gigas and Ostrea edulis and Their Possible Consequences for the Norovirus–HBGA Interaction"

_marinedrugs, 2023, doi:10.3390/md21060342_

Round 1

Reviewer 1 Report

 In the manuscript entitled “Species-specific N-glycomes and methylation patterns of oysters Crassostea gigas and Ostreaedulis and their possible consequences for norovirus-HBGA interaction” Guerardel et al defined the the tissue-specific N-glycomes of two oyster species, Crassostrea gigas and Ostrea edulis, showing how they syntetise different range of N-glycans,  that share similar terminal HBGA-like antigens, but carry different terminal residues and O-methylation pattern. The authors also show how the methylation pattern can tune the recognition of oysters by virus particles. The manuscript is well written, results reliable and well described. The manuscript can be accepted for publication in Marine Drugs

Author Response

We thank the reviewer for the kind assessment of our manuscript.

Reviewer 2 Report

The authors present a study on the N-glycoma of the oysters Crassostea gigas and Ostreaedulis, on the methylation pattern and on the possible consequences they may have in interactions with Noroviruses.

The research is done very well, with state-of-the-art methods and instruments and by very expert research groups. The subject is interesting, the manuscript written very well, the methods very well explained. The results are well illustrated and the conclusions sound reasonable. In my opinion the manuscript is worthy of publication in marine drugs.

Author Response

(The authors gave the same response as above.)

Reviewer 3 Report

The authors have submitted a very detailed and interesting study of the N-glycosylation profiles of two different oyster species. They show that the methylation patterns of terminal N-acetylgalactosamine and fucose residues of oyster N-glycans adds another layer of complexity to the already complex protein-linked glycosylation. This in itself is already worthy to be published.

In addition to the very nice and detailed data on oyster N-glycan profiles, the authors also present information on modelling of the interactions between norovirus capsid proteins and methylated oyster N-glycans. Yet, they correctly state that their modeling results should
be considered with caution since the models do not include effects of glycan clustering or the dynamic nature of virus-glycan interactions.

While the idea of norovirus enrichment in oysters is intriguing as is the potential interaction between methylated oyster N-glycans and norovirus capsid proteins, I feel that there is a stark contrast between the very detailed analytical N-glycan data and the rather speculative part about the modelling. The manuscript would benefit a lot from additional real wet-lab data on norovirus capsid interaction and methylated oyster glycans.

Another thing that I noticed: Some figures are cut at the top so that letters and words are affected. These figures should be replaced by the complete figures.

Author Response

We thank the reviewer for the useful comments. As mentioned in the review, the manuscript mainly focuses on the structural analysis of the oyster glycome and the identification of potential in vivo ligands for noroviruses. We strongly agree with the reviewer that the detailed analysis of the interaction mechanisms between methylated glycans and virus capsid is required to understand the fine specificity of VP1 with oyster carbohydrate ligands. However, such a study requires the use of a series of individually purified HBGA epitopes substituted with methyl groups at different positions, which are not currently available to obtain relevant data. We are currently undertaking a multidisciplinary approach involving (1) the chemical synthesis of three methylated human blood group A antigen glycans linked to an amine-terminated aglycon synthesised via a Galß(1,3)GlcNAc as a common building block, (2) the construction of a glycan array using synthetic and natural glycans probed with GI-3 and GII-4 VLPs, (3) the epitope mapping of synthetic and natural glycans by saturation transfer difference NMR using VP1 protein, (4) the far-western blot analysis of oyster glycoproteins using VLPs coupled to carbohydrate degradation enzymes, (5) ELISA experiments. Overall, we expect to publish in the coming months a comprehensive report on the fine specificity of norovirus towards oyster glycans in a forthcoming manuscript. However, this study appears to be beyond the scope of the present manuscript and we hope that the reviewer will agree with our view.

As recommended, we will provide a new set of high resolution figures that re not truncated in the final version of the manuscript.